# Effect of Geometric Curvature on Collective Cell Migration in Tortuous Microchannel Devices

**DOI:** 10.3390/mi11070659

**Published:** 2020-07-02

**Authors:** Mazlee Bin Mazalan, Mohamad Anis Bin Ramlan, Jennifer Hyunjong Shin, Toshiro Ohashi

**Affiliations:** 1Graduate School of Engineering, Hokkaido University, Sapporo 060-8628, Japan; mohamadanis77@gmail.com; 2AMBIENCE, School of Microelectronic Engineering, Universiti Malaysia Perlis, Arau 02600, Perlis, Malaysia; 3Department of Mechanical Engineering, Korea Advanced Institute of Science & Technology, Daejeon 34141, Korea; j_shin@kaist.ac.kr; 4Faculty of Engineering, Hokkaido University, Sapporo 060-8628, Japan; ohashi@eng.hokudai.ac.jp

**Keywords:** collective cell migration, tortuous microchannel devices, engineered tissue-scaffold

## Abstract

Collective cell migration is an essential phenomenon in many naturally occurring pathophysiological processes, as well as in tissue engineering applications. Cells in tissues and organs are known to sense chemical and mechanical signals from the microenvironment and collectively respond to these signals. For the last few decades, the effects of chemical signals such as growth factors and therapeutic agents on collective cell behaviors in the context of tissue engineering have been extensively studied, whereas those of the mechanical cues have only recently been investigated. The mechanical signals can be presented to the constituent cells in different forms, including topography, substrate stiffness, and geometrical constraint. With the recent advancement in microfabrication technology, researchers have gained the ability to manipulate the geometrical constraints by creating 3D structures to mimic the tissue microenvironment. In this study, we simulate the pore curvature as presented to the cells within 3D-engineered tissue-scaffolds by developing a device that features tortuous microchannels with geometric variations. We show that both cells at the front and rear respond to the varying radii of curvature and channel amplitude by altering the collective migratory behavior, including cell velocity, morphology, and turning angle. These findings provide insights into adaptive migration modes of collective cells to better understand the underlying mechanism of cell migration for optimization of the engineered tissue-scaffold design.

## 1. Introduction

More than two million surgeries are performed every year worldwide, requiring organ implants due to injuries or diseases. Current clinical treatment options for organ defects include autografts, allografts, synthetic organ replacements, and biodegradable engineered tissue-scaffolds [1,2]. Engineered tissue-scaffolds aim to replace the substantial loss in tissue or the entire organ to resolve organ shortage. However, fabricating an organ is not an easy task; it involves a thorough understanding of the underlying biology, as well as the ability to control many physiological factors [3]. Besides, it is necessary to understand the intimate cellular interactions with biomaterials used for scaffolds before designing a functionally suitable tissue constructs for an artificial organ.

Over the past two decades, significant efforts have focused on determining how an architecture of the engineered tissue scaffolds can affect cell behavior such as cell attachment, proliferation, differentiation, invasion, and colonization. In addition, collective behavior of cell migration has been recognized as an essential phenomenon within the designed tissue construct [4,5], as well as its natural significance in many pathophysiological processes such as wound healing, inflammation, angiogenesis, and cancer metastasis [3,6,7]. Furthermore, collective cell migration involves the coordination of mechanical and biochemical cues within a group of cells, which regulates cell-cell interactions and cell-environment communications [3,8,9,10,11,12]. However, most studies in tissue engineering had focused on attachment, growth, and proliferation of the constituent cells, neglecting the collective migratory behavior in response to the different cues from the microenvironments of the engineered tissue. Shen Ji et al. [13] investigated the effect of wavy scaffold architecture on the osteogenesis of human mesenchymal stem cell (hMSC) by ultilizing 3D porous scaffolds that featured curved or linear patterns. They found that hMSCs on wavy patterns spreaded by following the curvature form provided by the wavy patterns, exhibiting elongated morphology and mature focal adhesion points. The wavy pattern guided cells differentiated into the osteogenic lineage. Wismayer et al. [14] modified a tissue-engineered laryngeal scaffold’s nonporous surface membrane to allow successful cell entry. Their results indicated successful adhesion of porcine epithelial and p7072 porcine fibroblasts to the surface of the scaffolds, confirming that cells moved to the scaffolds. Pedraza et al. [15] fabricated a polydimethylsiloxane (PDMS)-based porous scaffold using particulate leaching (SCPL) technique, and they highlighted the importance of the extracellular matrix (ECM) coatings on the PDMS surface for cell attachment and cell proliferation. While the effects of different environmental factors have been tested on a variety of cells, the collective migratory behavior of cells has not been thoroughly investigated in literature.

In general, the engineered tissue scaffolds comprise of interconnected porous structures that warrant sufficient porosity for proper tissue integration while providing mechanical support and preserving tissue volume [16,17,18]. So far, many techniques, such as gas foaming, phase separation, electrospinning, and freeze drying, have been developed to optimize the tissue scaffold architectures [13]. However, not only it is difficult to control precisely a 3D structure of the scaffold, but also the imaging of the collective cell migration within a 3D structure remains as a great challenge [12,13]. Fortunately, the recent advancement in microfabrication techniques enables the investigation of collective cell migration under different microenvironments. Doran et al. [19] constructed a microfluidic device that allows examining the influence of varying channel widths, degrees of channel tortuosity, the presence of contractions or expansions, and channel junctions on the migration of mouse fibroblasts (NIH 3T3) and human bone marrow-derived mesenchymal stromal cell (hMSCs). Marel et al. [20] observed that the spreading cell sheet invaded the interstitial, channel-like areas of microstructures composed of Polyethylene glycol dimethacrylate (PEG-DMA) walls bordering the bare petri dish surface. Although these researchers studied the collective cell migration using 3D structure, how the geometric curvature from the sidewall boundary affects the collective cell migration in engineered scaffolds remains elusive. 

In this study, we developed a biocompatible tortuous microchannel device that mimics the various curvature of pores presented to the cells within 3D engineered tissue-scaffolds. The amplitude and radius of the curvature of the tortuous microchannel devices were varied to create four distinct pore tortuosity indices. A straight channel was used as a control. In both the control and tortuous microchannel devices, the width of the channels (pore size) was maintained at 50 µm. First, we confirmed the cell attachment at the edge of the tortuous microchannel devices. Then, we investigated the effects of contact guidance by the geometric curvature at the sidewall boundary to the collective cell speed, morphology, and distribution of the turning angle. The insights gained in this study will offer useful guidelines for designing and optimizing the engineered tissue-scaffolds.

## 2. Materials and Methods 

### 2.1. Design of Tortuous Microchannel Devices

The tortuous microchannel devices comprise of an inlet and an outlet with four different types of microchannels, as depicted in Figure 1a. In each of these devices, there are two sidewalls represented as the convex (positive curvature) area and concave (negative curvature) area. For the design of tortuous microchannel devices, we set the bending angle at 180° corresponding to the Triply Periodic Minimal Surfaces (TPMS) scaffold architecture [21]. The advantages of the TMPS scaffold architectures are that, it is easy and efficient to design a complex 3D porous scaffold model, and that, a variety of TPMS-based unit cell libraries can be generated to investigate the biomechanical and mass transport properties of tissue scaffold in the more systematic way [22]. We varied the amplitude and radius of the tortuous microchannel devices to provide a different of tortuosity index. The width and thickness of the tortuous microchannel devices (referring to pore size) are 50 µm × 50 µm to allow approximately 2–3 cell lengths in diameter, by considering that the width of the epithelial cell, which is in the range of 20–30 µm [23].

The details of the tortuous microchannel devices are described in Table 1. The tortuosity index is defined as the ratio of the actual pore length (segment length) to the linear distance between its ends (channel length) as shown in Equation (1) [24]. The tortuosity index of our tortuous microchannel varies from 1.57 to 2.30. The higher number of tortuosity index means the more twisted is in path in the channel.
(1)$$\mathrm{Tortuosity}\;\mathrm{index}=\frac{\mathrm{Segment}\;\mathrm{length}}{\mathrm{Channel}\;\mathrm{length}}$$

### 2.2. Preparation of Tortuous Microchannel Devices 

The fabrication process of the tortuous microchannels was carried out using photolithography and soft-lithography techniques [25]. Before the photolithography process, a chrome photomask was patterned by Electron Beam Lithography (EBL, ELS-3700, Ellionix Inc., Tokyo, Japan), as shown in Figure 1b. UV exposure and development were performed on a silicon wafer using negative photoresists, SU-8 3050, to produce a silicon mould. First, the SU-8 3050 was spin-coated on a silicon wafer to have a thickness of 50 µm; then, exposure was carried out using the patterned photomask and followed by the development process. After the development, the silicon mould was hard-baked inside the oven at 180 °C for 24 h. The surface of the silicon mould was salinized to minimize the adhesion effect.

The tortuous microchannel devices were constructed using a polydimethylsiloxane (PDMS, Sylgard 184 kit, Dow Silicones Corp., Midland, Michigan, USA). The PDMS is known to demonstrate biocompatibility and biostability following the clinical implantation [21,22]. Other benefits, include long shelf life, permeability to gas, transparency, and ability to observe cellular responses by inverted microscopy technique [19,20]. The PDMS elastomer base was mixed with the curing agent using a ratio of 10:1. The solution was mixed well for 5 min and then put inside a desiccator connected to the compressor pump for 30 min to remove bubbles. The mixture was then deposited on top of the silicon mould and baked at 110 °C for 20 min to provide the required stiffness of microchannel at approximately 489 kPa based on the compression test result using a digital push pull force gauge (DigiTech, AFK-500TE, Osaka, Japan). After it was cooled down, the tortuous microchannel devices were then peeled off carefully, and their edges were trimmed. The devices were cut, and holes were punched at the inlet and outlet of them.

Before cell seeding, the dimension of the tortuous microchannels was inspected using a digital microscope (VHX-600, Keyence Corp., Itasca, Illinois, USA). The tortuous microchannel devices were then sprayed with air duster to remove any dust and impurities. They were then cleaned in 70% ethanol using an ultrasonic cleaner for 1 min and then washed again for another 5 min in distilled water. The tortuous microchannels were dried under a clean bench and inspected under a microscope to ensure no dirt was attached. The tortuous microchannel devices and the cover glass to be attached (Matsunami Glass Ind. Ltd., Osaka, Japan) were sterilized under UV light for overnight inside a hood. Atmospheric plasma was applied to both tortuous microchannels and cover glass for 2 min to improve cell and immobilize soluble factors [26,27,28,29]. The tortuous microchannels were inverted and attached to the glass. To ensure firm attachment, a load, and heat at 100 °C were applied to the attached complex for 10 min. The firmness of attachment was tested by pulling the tortuous microchannels from the cover glass. To enable the cell attachment inside the tortuous microchannel devices, a small volume of fibronectin with a concentration of 50 µg/mL was introduced in the inlet of the tortuous microchannels. Fibronectin was aspirated into the channels through the outlet using a glass pipette ensuring it filled the channels. The setup was then left for 1 h to dry at room temperature. After an hour, a small amount of phosphate-buffered saline (PBS) was spotted in the inlet of the channels to rinse the fibronectin leftover. The PBS is aspirated until there was no liquid left on the setup to dry.

### 2.3. Cell Culture and Cell Seeding

Madin-Darby Canine Kidney cells (MDCK NBL-2, Cell Application Inc, San Diego, CA, USA) were used in this experiment as they exhibit the physiologic characteristics of the renal tubular epithelium [30]. The MDCK cells were cultured in Dulbecco’s Modified Eagle’s Medium (DMEM, Sigma-Aldrich, St Louis, Missouri, USA) supplemented with 10% Fetal Bovine Serum (FBS) and 1% penicillin and streptomycin (Wako, Japan) and maintained at 37 °C and 5% CO_2_ in the incubator. MDCK cells at passage 5 to 10 were used for all experiments.

The MDCK cells were seeded in the inlet, and the culture medium was applied in the outlet to avoid cells flowing into the tortuous microchannels before confluent state. The cell density was counted and maintained at 2 × 10^6^ cells/mL for all cell migration experiments. After 1 h, the culture medium was added to the cells to remove unattached cells from the inlet. Then the cells were cultured to confluent state overnight, in the incubator with an atmosphere of 5% CO_2_ at 37 °C. The experimental setup of the collective cell migration in the tortuous microchannels is shown in Figure 1c. Over 12 h of observation of collective cell migration, cell proliferation should be present, but, given that the MDCK population doubles at 21 h [31], the overall impact of proliferation on measured cell migration rates is negligible.

### 2.4. Time-Lapse Imaging

Once reaching a confluent state in the inlet, the collective cells started to move to the free edge inside the tortuous microchannels. The experimental setup was set in a mini-incubator (MIU-IBC-IF, Olympus Tokyo, Japan) to maintain the cells environment at 37 °C and 5% CO_2_ on an inverted microscope stage (IX81, Olympus, Tokyo, Japan). Time-lapse images were collected using an inverted microscope stage (IX81, Olympus, Tokyo, Japan) under a bright-field mode. The migration of the collective cells was tracked for 12 h after 24 h of cell seeding. The sequence of time-lapse images of the cell migration was obtained through the CCD camera (ORCA-R2, Hamamatsu Photonics, Japan). In order to get the entire image of the collective cell migration in the tortuous microchannel devices, we used a magnification of objective lens at 20× (LUCPLFN 20×, Olympus, Tokyo, Japan), and for the precise data the field delay was set at 10 min.

### 2.5. Visualization of Actin Filament and Nuclei

After observing collecting cell migration for 12 h, the MDCK cells were washed with PBS twice before being fixed using 4% paraformaldehyde in PBS for 30 min at room temperature. 0.2% Triton-X was added and the solution was left for 30 min. Rhodamine Phalloidin (Life Technologies) was introduced and the solution was stored inside a closed container at 4 °C overnight to stain the actin filaments. Cell nucleus was stained using Hoechst which was diluted with PBS at a ratio of 1:2000 introduced to the closed container and stored at 4 °C for 1 h. Finally, the cells were treated with prolong gold antifade overnight at 4 °C remaining in the closed container. After those treatments, cells were washed three times with PBS before start to do immunofluorescence microscopy took place. In order to see the details of the cortical actin filament, we used a 20× lens with oil on a Olympus IX81 inverted microscope. In order to visualize the actin cables of the front cells and to confirm the cells covered the inner surface of the tortuous microchannel devices, we use a confocal microscope (A1Rsi, Nikon Confocal System, Tokyo, Japan) with 20× objective lenses, which available in Nikon Imaging Centre (NIC), Hokkaido University, Japan.

### 2.6. Data Analysis

An image of collective cell migration in tortuous microchannel devices was recorded every 10 min using ImageJ software (Version 1.52i), in order to analyze the collective cell velocity, which was then measured from sequential time-lapse images based on the coordinate of the cells at leading-edge using Equation (2).
(2)$$\mathrm{Migration}\;\mathrm{velocity},\;v=\frac{\sqrt{{\left(x_{i+1}-x_{i}\right)}^{2}+{\left(y_{i+1}-y_{i}\right)}^{2}}}{t}$$
where (x_i_, y_i_) are the initial coordinate of cells (x_i+1_, y_i+1_) is the recorded cells track, and t is migration time in hours.

The magnitude velocity and vector field were mapped using Particle Image Velocimetry (PIV) technique. The purpose of the PIV analysis is to investigate the collective cell migration behaviour of cell monolayer. The PIV software we used was PIVlab, a MATLAB-based software, created by William [32]. The windows shifting technique was used, beginning with 4 or 64 wide images and finishing after three iterations with 16 or 16—the interrogation region overlapped by 50 per cent.

For the cell morphology, we analyzed the cell shape index (CSI) using circularity Equation (3) based on the position of the cells in the tortuous microchannel devices.
(3)$$\mathrm{Circularity}=\frac{2\pi A}{P^{2}}$$
where A and P are the area and perimeter of a cell, respectively.

We analyzed the cell orientation by observing the cell turning angle at the corner of the tortuous microchannel device using PIV. We used Equation (4) to measure the angle between the two vectors resulted from PIV analysis. Then, we estimated the turning frequency for 12 h of migration with an interval of 10 min. Before the analysis, we set the ROI by placing a mask at the convex and concave area.
(4)Turning angle=atan2(u,v)
where u= vector at x direction, v= vector at y-direction.

The cell orientation was divided into 3 different angle ranges; 0–30°, 30–60° and 60–90°. It was assumed that the cells were directed by the geometrical curvature when the cell orientation was within 30–60°.

### 2.7. Statistical Analysis

All statistical analysis was performed in GraphPad Prism 8 (GraphPad Software, San Diego, CA, USA). The data are presented as the mean ± standard deviation (SD), and the comparison between groups was carried out using a two-sample in Student’s t-test. The probability distribution function was used to test the normality of the data prior to Student’s t-test.

## 3. Results

### 3.1. Cells Adhered on The Fibronectin-Coated Sidewall and Not Adhered on Pluronic F127 Coated Sidewall

Cell attachment is an essential process in tissue scaffolds. Therefore, firstly, we investigated the cell adhesion at the bottom substrate and sidewall of the tortuous microchannel devices on the adhesive area. To do that, we applied 50 µg/mL of fibronectin inside the tortuous microchannel devices to create an adhesive surface. We observed that the front edge of the cell monolayer exhibited a concave-shaped structure perpendicularly aligned to the migration direction. This situation confirmed cell attachment, forming a meniscus, to the sidewall, as shown by a yellow arrow in Figure 2a. We also visualized the cortical actin filament of the cells in the tortuous microchannel devices as depicted in Figure 2b where the actin filaments at the boundary of the fibronectin coated surface.

### 3.2. MDCK Cell Migrate Collectively to Maintain the Cellular Integrity in Tortuous Microchannel Devices

After confirming the cell attachment at the substrate and sidewall of the tortuous microchannel devices, we observed the characteristics of MDCK cell migration for 12 h. Our results indicated that MDCK cells contacted the sidewall and moved coherently throughout the tortuous microchannel devices, as depicted in Figure 3a. We also viewed the cross-section of the MDCK cells monolayer, and observed that the epithelial cells covered the inner surface including the bottom substrate and sidewalls of tortuous microchannel devices, as shown in Figure 3b.

### 3.3. The Influence of Radius and Amplitude of Tortuous Microchannel Devices on The Cells’ Front Velocity

After confirming the formation of the MDCK cell sheet, we examined the velocity of collective cell migration in the tortuous microchannels using a time-lapse video as shown in Appendix A in the Appendix A. We tracked the migration distance of the leading edge of cell sheet up to 12 h of observation time. We found that the migration distance increased monotonically with the increase in time, as shown in Figure 4a. Then, we quantified the average velocity of the cells at the leading edge in all channels; 39.2 ± 5 µm/h (control), 37.6 ± 6 µm/h (channel 1), 47.7 ± 9 µm/h (channel 2), 37.1 ± 11 µm/h (channel 3) and 49.6 ± 20 µm/h (channel 4). We discovered that the channels of higher radius and channel amplitude significantly exhibited higher cell migration velocity, as shown in Figure 4b. To understand this phenomenon, we sought the dominant factor that influenced the collective cell migration velocity. To achieve that, we divided the tortuous microchannels into three regions; regions I, II, and III, as shown in Figure 4c. We assumed areas I and III comprised of higher curvature than region II. Then, we examined the average migration velocity based on the stated area.

Interestingly, we observed that a different velocity rate occurred within the tortuous microchannels, as depicted in Figure 4d. Our results showed that MDCK cells accelerated when entering the region II. After the leader cells have entered the area I, they decelerated before cells enter the next region.

### 3.4. The Effect of Concave and Convex Sidewall Curvature on Cell Velocity of The Cell Monolayer

To further understand the effects of the curve at the boundary sidewall, we used PIVlab software to analyze the velocity magnitude of the cell monolayer in the tortuous microchannel devices. The PIV analysis indicated that the MDCK cells showed a coordinated migration with the highest speed detected for cells at the front of the monolayer in both straight microchannel and tortuous microchannel devices (Figure 5a and Appendix A in the Appendix A). We observed that majority of the cells in the tortuous microchannel devices moved slower at the corner of the tortuous microchannel devices, as depicted in the yellow arrow in Figure 5a. While in the straight area, cells accelerated their speeds in all channel architectures examined. As shown in the previous section, the high curvature region affected the average velocity of the collective cell in the tortuous microchannel devices. Therefore, we further analyzed the area mean value of the velocity magnitude of cells at the concave and convex area. Before starting the PIV simulation, we set the region of interest (ROI) by applying a mask near the sidewall boundary. Then, we compared the results of cell speed at the border and middle of the straight microchannel. Our results indicated that cells at the sidewall of the straight microchannel moved approximately 2.5-fold faster than those at the concave and convex area in the tortuous microchannels as shown in Figure 5b. Furthermore, the cells at the convex area moved faster than those in the concave area, whereas the average velocity in the straight microchannel was about the same at both borders. The cells in the channel 1 and 2 exhibited quite a similar velocity magnitude due to the same radius (25 um), and similar observation was also made in channel 3 and 4. We also compared the cells velocity between leader and follower cells. Our results showed that the leader cells exhibited higher speed than follower cells. Interestingly, we found the follower cells speed were similar in all tortuous microchannel devices as shown in Figure 5c.

### 3.5. The Effect of Concave and Convex Sidewall Curvature on Cell Morphology

To further examine the effects of the geometric curvature on the collective cell behaviors, we compared the morphology of cells in a straight microchannel versus tortuous microchannel devices. To achieve that, we stained the MDCK cells with Rhodamine Phalloidin and Hoechst to visualize the actin cytoskeleton and nucleus, respectively, after 12 h of migration as shown in Figure 6a. We could quickly identify the MDCK cells based on their high intensity of cortical actin signal bordering each other. To quantify the cell morphology systematically, we divided the area of the straight microchannel into two main areas; border and middle. While, in the tortuous microchannel devices, we divided the tortuous microchannel into three areas; concave, middle, and convex. Using the CSI analysis, we found that cells were significantly elongated at the sidewall boundary compared to ones in the middle area of the straight and tortuous microchannel devices (Figure 6a,b). Interestingly, we observed the cells at leading-edge forming a concave curvature by rearranging the actin bundles in an orientation perpendicular to the free cell edge in the tortuous microchannels, as shown in Figure 6c.

### 3.6. Turning Angle of Collective Cells Deviated at Boundary Curvature

The relative turning angle describes how much a track deviates from its previous direction, and is also used to measure the persistence [33]. We have hypothesized that the collective cells may deviated at the turning point of the tortuous microchannel devices. Therefore, we analyzed the turning angle distribution of the MDCK cells to further examine the effect of boundary curvature on collective cell migration. We extracted a vector direction in degrees by selecting the area at the corner of the tortuous microchannel using PIVlab as shown in Figure 7a. Then, we divided vector direction into 3 different angle ranges; 0–30°, 30–60° and 60–90° as illustrated in Figure 7b. We assumed the cells with migrated towards 30–60° are directed by the curvature, while, the cells were deviated when their turning angle are 0–30° and 60–90°. As a result, the collective cells are more deviated from their direction in the channels with lower tortuosity (channels 1 and 3) as compared to cells in the channels with higher tortuosity (channels 2 and 4) in Figure 7c. Consequently, the cells which deviated at the turning point of the tortuous microchannel devices may decrease the velocity of the collective cells.

## 4. Discussion

Development and regeneration of cells in engineered tissue scaffolds require the coordination of collective cells through the various interactions with the surrounding environment and cell-cell communication. Many studies so far have demonstrated the effects of local geometric cues on cell attachment, proliferation, and differentiation within the engineered tissue scaffolds, but the migratory behavior of collective cells within the scaffolds remains elusive. In this study, we fabricated PDMS-based tortuous microchannels to explore the collective cell migration under different geometric constraints that mimic the porous architecture with various curvatures in engineered scaffolds. One of the common issues encountered in the engineered tissue scaffolds is the cell attachment on the edge of the scaffold. We revealed that the cells at the leading edge of the cell monolayer formed a plug-flow-like profile on Pluronic F127 coated sidewalls. The observation had an agreement with Marel et al. [20], where they applied Polyethylene glycol dimethacrylate (PEG-DMA) to avoid cell adhesion at the wall. In this study, we confirmed the cell attachment in the inner surface of the tortuous microchannel devices, thus mimicking the physiological condition in scaffolds used in tissue engineering applications.

Our results indicated a transition of migratory patterns in response to the different geometric curvatures during collective epithelial cell migration. We found that varying the radius and amplitude of the tortuous microchannel (tortuousity index) resulted in a distinct migration velocity of collective cells. These results were in contrast to the previous study done by Mills et al. [34], as they observed that there were no variations in cell front migration rate of 3T3s and hMSCs within the tortuous channel whose tortuosity index was in the range of 1.1 to 2.2. The discrepancy might have been mainly due to the fact that the channel width of 200 µm was 4 times larger than the tortuous microchannel device. Besides, the model developed by Mills et al. was based on 2D while we implemented 3D constricted area to observe collective cell migration affected by geometric curvature. Jain et al. [35] performed cell migration experiments using rings with different width at 20, 50, 100 and 200 µm. They found that the global rotational movements were independent of geometrical constraints. This observation was similarly reported in the paper by Chen et al. [36]. They used a wound healing assay to observe the actin organization at the leading edge of MDCK cells. They found that the cells at the front edge displayed with a concave shape (negative curvature) showing actin cable assembly. However, in our study, we maintained the width at 50 µm and we observed that cells were encountered with convex and concave curvature throughout the tortuous microchannel device. The observation that migration was enhanced in tortuous as compared to straight microchannels as shown in Figure 4a,b was very interesting. This phenomenon may be due to the better organization of cells at the leading edge in the tortuous microchannel device as compared to the straight microchannel, consistent with what was reported by Vedula et al. [37]. Vedula et al. reported that cells in straight microchannel of 20 µm in width exhibited significantly negative velocities whereas the large velocities were observed everywhere across the whole length of cell chain not necessary in the leading edge. In the straight microchannel, the cells at the leading edge were less directed as compared to the ones in the tortuous microchannel device. This observation also was in agreement with Young et al. [38] where the authors reported that the straight microchannel showed no observable directional preference in any direction.

We suggested that contact guidance of the sidewall curvature must play an important role on the collective cell migration in engineered tissue scaffolds. Stachowiak et al. [39] reported that the contact guidance influenced the motility behavior of T cell through the collagen coated poly(ethylene) hydrogel scaffolds. To test the hypothesis, firstly, we coated the inner surface of the tortuous microchannel devices with the fibronectin as the ECM, including the bottom substrate and sidewalls. Then, we validated the cell adhesion by examining the intensity of actin filaments at the curvature boundaries. Our results confirmed that the cells at the leading edge consisted of one or more cells formed a concave surface, perpendicular to the free cell edge in the tortuous microchannel devices due to the contact guidance at the sidewall. The hypothesis also supported by Tarle et al. [40] where they observed that the concavity at the front edge in their collective modelling under geometric confinement reinforced a pull of the cells towards the adhesive boundary. Using the PIV analysis, we also quantified the velocity of cell monolayer affected by the contact guidance at the concave and convex area, confirming that boundary curvature influenced both leader and follower cells. We showed that the leader cells with the concave-shaped structure lead the cell migration throughout the tortuous microchannel devices. This concavity at the front cells reminisced the wound healing process, where a multicellular actin belt consisted of actin stress fibers that lined the perimeter of cell collectives in a purse-string mode of motility [41]. These observations suggested that the purse-string–like movement occurred at the leading edge and contributed to propelling the cell monolayer forward during cell development and regeneration in tissue scaffolds. This observation was consistent with the report by Chen et al. [36] where the authors used a wound healing assay to observe the actin organization at the leading edge of MDCK cells. They found that the cells at the front edge displayed with a concave shape (negative curvature) showing actin cable assembly.

In this study, one of the main findings was that the MDCK cells moved slower at the curved areas as compared to the movement of the cells at the straight path. Our hypothesis suggested that the leader cells directed the movement pattern by sensing and responding to the local cues of the surrounding microenvironment. The interactions between the neighboring cells was critical to collective cell migration at the curvature area. During the collective cell migration, we observed that cells migrated coherently while maintaining the cell-cell contacts in the tortuous microchannel devices. Kevin et al. [42] reported that high cell cohesiveness was often associated with a reduction in speed as the inhibition of adhesive proteins lead to an acceleration of migration. Also, the other researcher observed that epithelial cells would slide past each other and change neighbors when moving collectively and cohesively [43]. The orchestration of intimate cell-cell interactions and leader cell formation would cooperatively direct the rest of the cohort [44].

Numerous microdevices have been developed in order to study the effects of the physical cues, including substrate stiffness and topography, to improve cell colonization within the engineered tissue scaffolds. For instance, Shen Ji et al. [13] constructed 3D printed scaffolds, and showed that hMSCs attached to wavy patterns by spreading conformally along the curvatures provided by the wavy patterns with elongated morphology. The results were similar to the present study where we observed the low circularity of MDCK cells at the concave and convex area. However, in Shen ji et al.,’s study, the pore size used was approximately 350 µm which was slightly bigger than the optimal pore size reported for significant cell growth, ranging from 100–135 µm [45]. Herein, we developed a smaller pore size at 50 µm width, and interestingly, we found the excellent cell attachment and coherent collective migration in the tortuous microchannel. Thus, this result serves as a guideline for improving the cell colonization in engineered tissue scaffolds.

The tortuous microchannel devices influenced the orientation of the actin bundles. The actin bundles were aligned along the sidewall of the curved surface in perpendicular to the cell migration path at the leading edge of the cell monolayer. This observed phenomenon was similar to many previous studies. For example, Soarez et al. [46] observed that actin bundles were mainly aligned along the long axis in pill-shaped chambers with a length/width ratio of 2 or higher. They suggested that confinement-induced actin bundling was guided with each other and the walls by steric repulsions from the filaments. This phenomenon showed that the physical mechanism from the sidewall curvature, in conjunction with adhesion, could influence the intracellular organization of actin filaments [47].

After observing through PIV that the collective cells moved slower near to the curvature, we explored the turning angle at the corners of the tortuous microchannel devices to understand a possible mechanistic explanation for our initially intuitive observation of the motility of cells. We demonstrated that the proportions of the MDCK cells in the 30–60° angle ranges were larger than the other angle ranges in all tortuous microchannel devices. This result suggested that the cells were guided along the curvature direction, consistently with the existing literature by Shijie et al. [48]. Interestingly, we observed that the cells in the tortuous microchannel with higher tortuosity (channel 2 and 4) were more highly oriented as compared to the cells in the lower tortuosity microchannels (channel 1 and 3). These results may prove that the cells with good orientation with the geometric curvature, migrate faster in the tortuous microchannel devices. The deviation of the turning angle may influence on the velocity magnitude of cells at the leading edge and cell monolayer and it also may be due to collision in locomotion (CIL) occurred at the turn of sidewall curvature [26] during collective cell migration. Another mechanism that may induce the decrease in cellular speed at the curvature is the enhanced cell proliferation and cell growth. Bidan et al. [49] reported that by merely tuning the curve of the surfaces where the cells attached their extracellular matrix would promote tissue growth. A simple geometric model based on the cell’s tensile behavior, which lead to curvature-controlled growth, could predict both the kinetics achieved and the tissue deposition distribution. Therefore, in this study, the tension between cells at the bottom substrate and sidewall might have induced cell proliferation and cell growth, affecting the collective cell migration in the tortuous microchannel devices. However, this hypothesis needs to be investigated further in future studies.

## 5. Conclusions

In conclusion, we developed tortuous microchannel devices by mimicking the geometric of the pore curvature as being presented to the cells within 3D engineered tissue scaffolds. This study highlights a systematic investigation to reveal the underlying mechanism of collective cell migration in the tortuous microchannel devices. Indeed, the geometric curve from the sidewall boundary of these tortuous microchannel devices influenced the group cell velocity, morphology, and turning angle at specific space and time in tortuous microchannel devices. The findings of the present study conclude that the tortuosity index obtained by manipulating the radius and amplitude of the curvature plays a vital role in the cell movement within the tortuous microchannel devices. These findings provide us with an insight on how collective cell migration behaves in the engineered tissue scaffold, and they can be useful for designing of tissue-based substrates.

## Figures and Tables

**Figure 1 micromachines-11-00659-f001:**
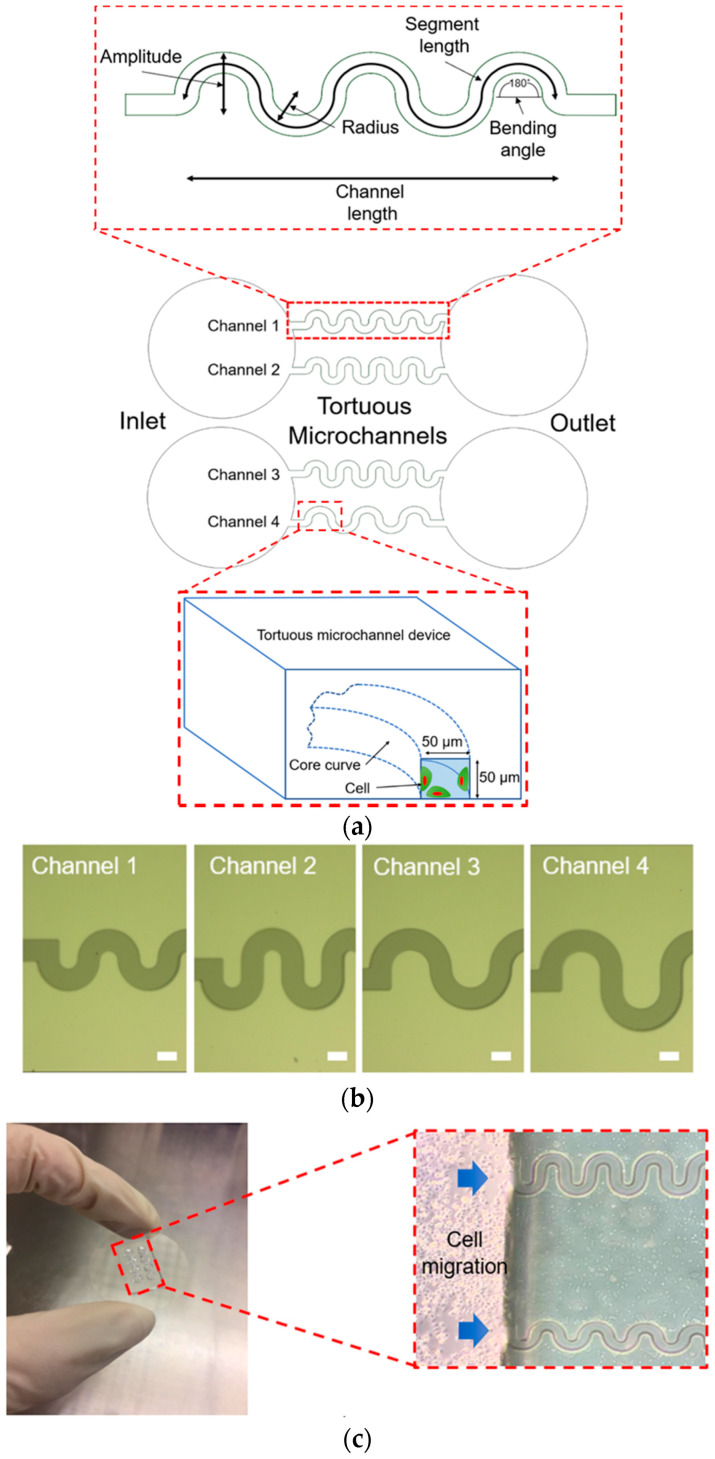
(**a**) Schematic diagram of the tortuous microchannel devices and illustration of cells inside the tortuous microchannel. (**b**) The patterned photomask of tortuous microchannel device. (**c**) The tortuous microchannel devices are mounted on the cover glass; the inset shows the experimental setup (Scale bars: 50 µm).

**Figure 2 micromachines-11-00659-f002:**
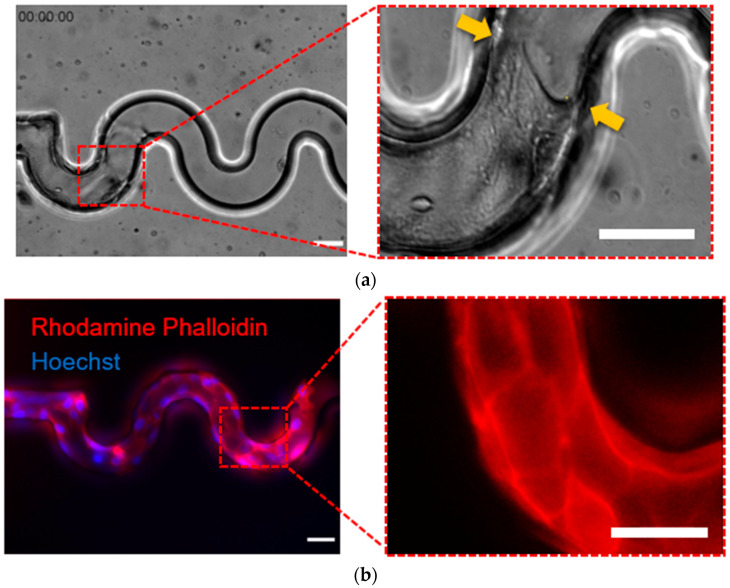
(**a**) Sidewall coated with fibronectin coated presented in the phase-contrast image. (**b**) The fluorescence image shows the actin filament and nucleus (stained by Rhodamine Phalloidin and Hoechst, respectively) of cells in a tortuous microchannel device. (Scale bar: 50 µm).

**Figure 3 micromachines-11-00659-f003:**
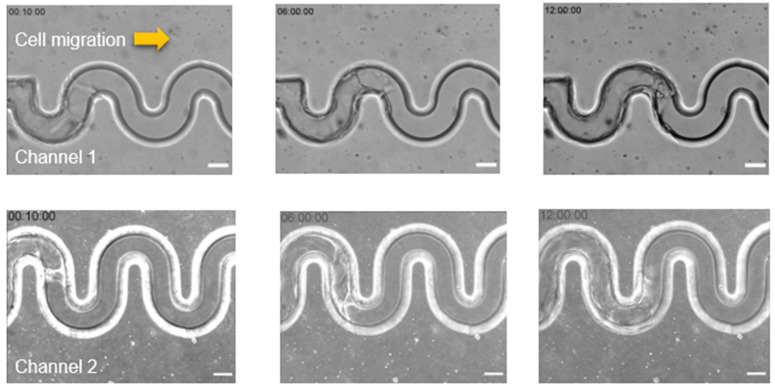
(**a**) Representative time-lapse images are showing collective cell migration in different tortuous microchannels at 10 min, 6 h and 12 h of cell migration. The yellow arrows indicate the migration direction (**b**) Top and cross-section views of collective cell in a tortuous microchannel device showed the cells covering the inner surface of the channel (Scale bar: 50 µm).

**Figure 4 micromachines-11-00659-f004:**
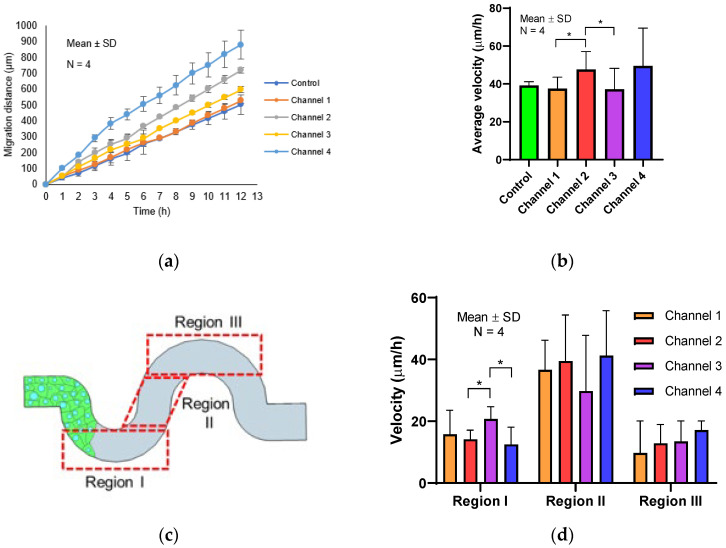
(**a**) Migration distance of Madin-Darby Canine Kidney (MDCK)cells front in different tortuous microchannels, (**b**) Average velocity of cells front in control and different tortuous microchannels, (**c**) Illustration of a tortuous microchannel with three different regions of interest. (**d**) The average velocity of cells front in different areas. Asterisk (*), *p* < 0.05 as obtained by Student’s t-test statistical analysis (N = 4 independent experiments in each condition).

**Figure 5 micromachines-11-00659-f005:**
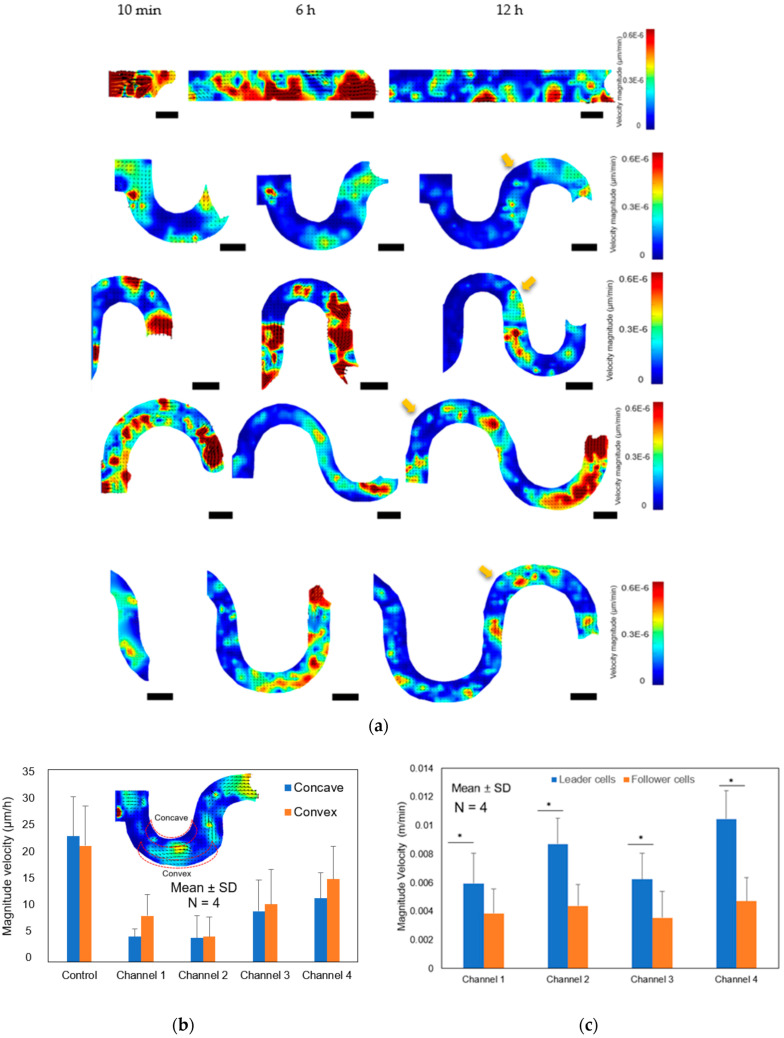
(**a**) PIV analysis on magnitude velocity and vector in different geometric curvature. (**b**) Average of magnitude velocity of straight and tortuous microchannel **(c)** Comparison of magnitude velocity between leader and follower cells in the tortuous microchannel device. Asterisk (*), *p* < 0.05 by Student’s t-test statistical analysis (N = 4 independent experiments in each condition, scale bar: 50 µm).

**Figure 6 micromachines-11-00659-f006:**
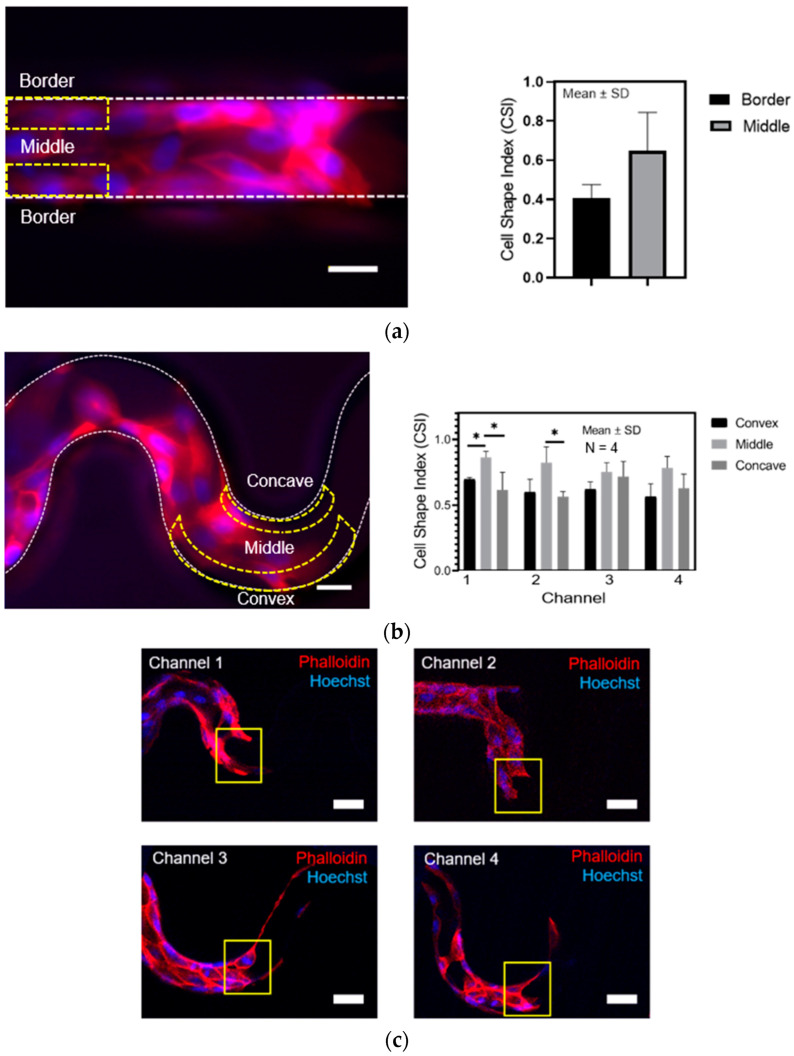
Comparison of (**a**) straight and (**b**) tortuous microchannel based on magnitude velocity and cell circularity. (**c**) The leader cells formed a concave-shape. Asterisk (*), *p* < 0.05 by Student’s t-test statistical analysis (N = 4 independent experiments in each condition, scale bar: 50 µm).

**Figure 7 micromachines-11-00659-f007:**
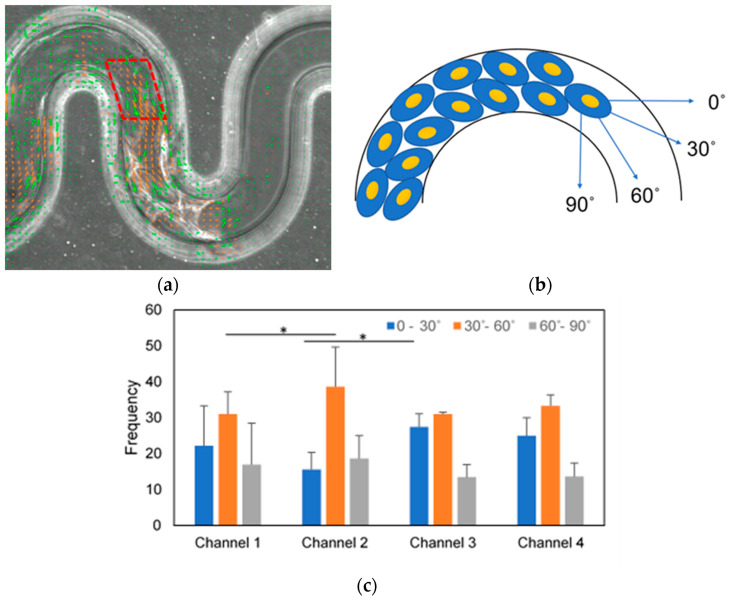
(**a**) Cell alignment in tortuous microchannel devices. (**b**) Cell vector directions (green and orange arrows) and ROI in a red dotted parallelogram. (**c**) Histograms of the statistics of cell direction for three angle ranges, inset is the schematic of turning angle of cell alignment. Asterisk *), *p* < 0.05 by Student’s t-test statistical analysis (N = 4 independent experiments in each condition, scale bar: 50 µm).

**Table 1 micromachines-11-00659-t001:** Dimension and tortuosity index of the microchannels.

Channel	1	2	3	4
**Radius (µm)**	25	25	50	50
**Amplitude (µm)**	75	100	100	125
**Segment length (µm)**	157	207	236	286
**Channel length (µm)**	100	100	150	150
**Tortuosity index**	1.57	2.3	1.57	1.91

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
