# Peer review of "Effect of Geometric Curvature on Collective Cell Migration in Tortuous Microchannel Devices"

_micromachines, 2020, doi:10.3390/mi11070659_

Round 1

Reviewer 1 Report

The manuscript is well written, and it describes and characterizes migration phenomenon is a new and interesting way to quantify effects of frequency and length of channel curvature and angle of channel curvature on rates of cell migration. the model could also serve tissue engineering tools for other cells/tissue types or as a potential screening tool for therapeutics for migrating tumor cells.

There are few shortfalls which must be addressed prior to further consideration.

1) mechanics are mentioned early on as playing a role in migration. However, it seemed that mechanical stiffness of substrates were uniform in the PDMS structures. Were they not measured or theoretically calculated for the different channel types?

2) the description of pluronic addition should be in the methods next to fibronectin coating.

3) why were human vascular endothelial cells tested, which seems to be more appropriate?

4) were other substrates tested, such as gelatin, collagen or even poly-l lysine? discuss, since it could open the model to other cell/tissue types

5) the biggest concern with the study is how did the authors ensure that the cells were actually migrating? The results could be confounded by proliferating cells. one could show staining of known migration markers? alternately staining of proliferation markers could show how little proliferation there is. what is the doubling time for this kidney cells? counting of DAPI cells is start.  

Author Response

Dear Sirs,

Best regards,

Mazlee Bin Mazalan

PhD candidate

Laboratory of Micro-Biomechanics

Hokkaido University

Reviewer 2 Report

Overview

In this manuscript, Bin Mazalan et al. develop tortuous microchannel devices to study collective cell migration in vitro. They propose that such devices enable in vitro characterisation and study of collective cell motion with a specific emphasis on understanding the effects of geometric curvature. How extracellular curvature affects collective cell migration has been an overlooked concept in the field. Bin Mazalan and colleagues use the tool to characterise the collective migration of MDCK cells.

The development of the tortuous microchannels they use is certainly a useful tool to address the role of curvature on collective cell migration. I also have no qualms with the idea of simply characterising collective motion in such microchannels as an initial study. However, I have major concerns regarding the quality of the imaging, presentation of data, quality of written English and contextualisation of results. These would all need to be addressed before acceptance of the manuscript. If these can be addressed to improve the quality of the manuscript, then this will be a very nice paper to have published.

Major Concerns

  1. The authors use bright field images in Figs. 2A and B to claim attachment or lack of attachment to the substrate. However, although to my eyes the cells do appear adhered in the fibronectin but not plutonic case, this is not sufficient to draw this conclusion. The authors must show adhesion to the extracellular matrix substrate, for example, by showing evidence of focal adhesions on the fibronectin but not on the Pluronic. This could be done in any number of ways e.g. immunostaining of any number of appropriate components.
  2. In the main text, there is no description at all as to what is labelling the actin in Fig. 2C, not even in the figure legend. Such information must be stated.
  3. The graph of Fig. 2d is poorly presented. The symbols of Pluronic and Fibronectin are far too similar which makes understanding the graph difficult. These should be changed to two different colours for easier distinction. Also, error bars should be included on each after doing calculations from a number of cell junctions. Showing just one is insufficient to make the conclusion that actin has stronger intensity on fibronectin sidewalls compared to Pluronic. Also, in Fig. 2E, the panels should be ordered in the same way as Fig. 2A-C i.e. fibronectin then pluronic. They should also be labelled appropriately (Fn or Pluronic) on the diagram and a scale bar included. Importantly, the authors must show representative pictures. The image of actin in Fibronectin at the sidewall looks completely different in C compared to E. In fact, the size of the fluorescence staining in C makes me think it is not cortical at all – the cell cortex is only 100-1000 nm thick. The significance of actin at the edge is also unclear – is it being shown as validation of successful adherence (if so, why?) or to illustrate something else?
  4. 3A claims to show collective cell migration. Yet it uses single frames, which is insufficient, particularly when using a yellow arrow to claim collective migration. The frame time stamp is (presumably) 2 hours (this timing notation must be explained in the text) which indicates the authors have imaging up until that point. Ironically, video 1 shows collective migration and yet this is not even referenced in the text. This video (and video 2) must be referenced appropriately and Fig. 3A must contain multiple frames of a time lapse movie to illustrate collective migration. The same must be done for each of the four channels. Cell tracks would also be useful (available on any simple ImageJ plugin like TrackMate or Manual Tracking).
  5. The authors say that Fig. 3B is evidence of cells covering the bottom of the channel, yet the side view pictures do not show where the bottom of the channel lays. It should be included.
  6. The authors say that channels 2 and 4 have the highest velocity. But channel 4’s variation is so big that it is not statistically significant. The authors must be clear about this and explain why channel 4 is not faster based on the parameters they are suggesting are important for collective migration in their microchannels.
  7. The authors divide the microchannel into 3 regions (Fig. 4C), claiming they can “assume areas I and II comprise of higher curvature”. The authors do not need to make such assumptions. The microchannels are made themselves, and they have access to all the geometric information. The authors should quantify the curvature in each region.
  8. The English is of poor quality throughout the text. This absolutely must be amended before acceptance and publication. There are too many cases to name them all, so I recommend getting it checked by a native or expert English speaker. In one case (line 326-7) this leads to ambiguous or incorrect conclusions or descriptions based on the data. In this case, the cells do not accelerate to maximum speed when entering the region II – there is no evidence of this. Instead, simply that cells are maximum speed when in region II.
  9. For graphs Fig. 4a, b and d, colour should be used rather than shades of grey or symbols because they are very difficult to interpret. In all figure legends where appropriate, n numbers (number of cells) should be included as well as N numbers (number of experimental repeats). For Fig. 4d in particularly, it is important to state whether this is only quantification of leaders or also includes followers. A graph of both leaders and a graph of followers should be quantified to see whether differences in curvature are affecting leaders and followers differently.
  10. The observation that migration distances are enhanced in tortuous compared to straight microchannels (Fig. 4A, B) is very interesting but not followed up on or explained in anyway. It also does not fit with the PIV data (Fig. 5), which show in the straight microchannels of cells having a high velocity, whereas in the tortuous channels (which clearly have migrated further) overall the velocity seems comparably reduced. These results need explaining – how can cells in straight channel move faster but travel less distance?
  11. There should be quantification (a graph) of the result described on line 351-353
  12. Why does region 1 show significant differences but not region III (in fig 4d)? These have identical (albeit opposite) curvatures. The explanation and interpretation of this graph is totally unclear.
  13. 5A – the scale bar and panels are totally unlabelled – this must be fixed. The authors say the front of the monolayer is the fastest moving region but there is no quantification of this. It is also unclear whether the front red (fast moving) regions correspond to one row or cells or multiple rows.
  14. Higher magnification pictures of Fig. 6C would be much better as it is difficult to see the so-called perpendicular actin cables the authors suggest is at the leading edge. They should also show separated and merged channels to make this easier to see. These images are not convincing.
  15. The significance of the sentence starting on line 441 is unclear. Also, factually speaking, the data to which it refers does not even quantify this difference. The angle plots are not sufficient to obviously show what the authors claim. Moreover, this figure is highly confusing and should be remade. The direction of angle alignment in the panels is unclear (is up/zero supposed to be straight/no turning angle)? Channel 3 and 1 may show slightly more deviation than 2 and 4 but this should e quantified. The context of this result is unclear and how it relates to alignment in a straight channel is unclear (Fig. 7H). Additionally, I am unclear how these results relate to the velocity and shape indices measured previously in the manuscript.
  16. Following on from the last point, a final panel in the manuscript which describes the parameters measured (velocity, persistence, cell shape etc) in different regions of tortuous microchannels vs straight microchannels would be useful, if the authors are claiming (or trying to show) some relationship. The use of the arbitrary “channel 1”, “channel 2” etc. makes comparisons for the reader extremely confusing (see minor concern 1).
  17. There are some key papers that the authors should reference and discuss. Jain et al., 2020 (Nature Physics) uses curved microchannels with different geometries; Pieuchot et al., 2018 (Nature Communications) demonstrate directed migration by curved topology. Chen et al., 2019 (Nature Physics) shows curvature sensing by directional actin flows during cell migration. These major papers and others should be references and discussed with regard to how they fit with your results. (e.g. the latter paper shows negative curvature induces polarised actin formation, similar to what you see in your model).

Minor Concerns

  1. Reference to “channel 1”, “channel 2” etc. is inappropriate. The authors should name these channels in accordance with their characteristics. As it stands, the figures that include “channel 1” etc are not self-explanatory and require the reader continually referencing Figure 1 to understand what is being used.
  2. In Fig. 2A, the yellow arrow is practically redundant; it is not even pointing to the sidewall which is what is claimed in the main text.
  3. 2A/B/C – for the sake of clarity, scale bars on the zoom out and inset panels should represent the same distance.
  4. 4A – change symbols for colours as the graph is difficult to read.
  5. 7B – what are the orange arrows? A zoom showing the arrows would be useful as it is difficult to see.
  6. It is unclear (or at least not stated) whether normality tests were carried out before using two-sample student’s T-test as a statistical test, as it was used in all stats throughout the manuscript.
  7. The alignment method must be described and explained in the methods and also briefly in the main text as it is not clear how this was done and consequently what the figure is trying to show.
  8. The authors should expand and be more specific as to the potential reasons their results conflict with a published study (lines 489-491).
  9. I do not believe the suggestion the cells behave like a purse string during this type of migration. There is no evidence for that other that the (dubious) imaging suggesting an actin cable at the front. Apart from lack of evidence that this actin cable is contractile and any functional tests, cables have been suggested for various things in collective migration (Shellard & Mayor, JCS 2019).

Author Response

(The authors gave the same response as above.)

Reviewer 3 Report

A very solid study about how the geometric cues (here the curvature) impact the collective cell migration. Especially, I appreciate the design of such devices, very smart idea. My only major concern is that if the authors can do one additional experiment on inhibiting the cell contractility (such as the treatment of Blebbistatin or Y-27632). Previous study has shown that the vortex disappeared in the cells with reduced contraction (10.1073/pnas.1119313109). It would be interesting to see if the geometric cue is eventually translated into such contraction-driven mechanical cue as well.

There are two small comments. 1) If the authors can add one plot of velocity alignment between cells in different regions. I expect that the alignment would drop dramatically in the region with high curvature.
2) Please add the following references on mechanical cues in cell collective migration to the introduction part:
10.1016/j.bpj.2019.05.020
10.1007/978-3-030-17593-1_4
10.1007/s10237-020-01308-5

Author Response

(The authors gave the same response as above.)
